# An Enhanced Scheme for Reducing the Complexity of Pointwise Convolutions in CNNs for Image Classification Based on Interleaved Grouped Filters without Divisibility Constraints

**DOI:** 10.3390/e24091264

**Published:** 2022-09-08

**Authors:** Joao Paulo Schwarz Schuler, Santiago Romani Also, Domenec Puig, Hatem Rashwan, Mohamed Abdel-Nasser

**Affiliations:** 1Departament d’Enginyeria Informatica i Matemátiques, Universitat Rovira i Virgili, 43007 Tarragona, Spain; 2Electronics and Communication Engineering Section, Electrical Engineering Department, Aswan University, Aswan 81528, Egypt

**Keywords:** EfficientNet, deep learning, computer vision, image classification, convolutional neural network, DCNN, grouped convolution, pointwise convolution, data analysis, network optimization, parameter reduction, parallel branches, channel interleaving

## Abstract

In image classification with Deep Convolutional Neural Networks (DCNNs), the number of parameters in pointwise convolutions rapidly grows due to the multiplication of the number of filters by the number of input channels that come from the previous layer. Existing studies demonstrated that a subnetwork can replace pointwise convolutional layers with significantly fewer parameters and fewer floating-point computations, while maintaining the learning capacity. In this paper, we propose an improved scheme for reducing the complexity of pointwise convolutions in DCNNs for image classification based on interleaved grouped filters without divisibility constraints. The proposed scheme utilizes grouped pointwise convolutions, in which each group processes a fraction of the input channels. It requires a number of channels per group as a hyperparameter Ch. The subnetwork of the proposed scheme contains two consecutive convolutional layers K and L, connected by an interleaving layer in the middle, and summed at the end. The number of groups of filters and filters per group for layers K and L is determined by exact divisions of the original number of input channels and filters by Ch. If the divisions were not exact, the original layer could not be substituted. In this paper, we refine the previous algorithm so that input channels are replicated and groups can have different numbers of filters to cope with non exact divisibility situations. Thus, the proposed scheme further reduces the number of floating-point computations (11%) and trainable parameters (10%) achieved by the previous method. We tested our optimization on an EfficientNet-B0 as a baseline architecture and made classification tests on the CIFAR-10, Colorectal Cancer Histology, and Malaria datasets. For each dataset, our optimization achieves a saving of 76%, 89%, and 91% of the number of trainable parameters of EfficientNet-B0, while keeping its test classification accuracy.

## 1. Introduction

In 2012, Krizhevsky et al. [1] reported a breakthrough in the ImageNet Large Scale Visual Recognition Challenge [2] using their AlexNet architecture, which contains 5 convolutional layers and 3 dense layers. Since 2012, many other architectures have been introduced, like ZFNet [3], VGG [4], GoogLeNet [5], ResNet [6] and DenseNet [7]. Since the number of layers of proposed convolutional neural networks has increased from 5 to more than 200, those models are usually referred to as “Deep Learning” or DCNN.

In 2013, Min Lin et al. introduced the Network in Network architecture (NiN) [8]. It has 3 spatial convolutional layers with 192 filters, separated by pairs of pointwise convolutional layers. These pointwise convolutions enable the architecture to learn patterns without the computational cost of a spatial convolution. In 2016, ResNet [6] was introduced. Following VGG [4], all ResNet spatial filters have 3 × 3 pixels. Their paper conjectures that deeper CNNs have exponentially low convergence rates. To deal with this problem, they introduced skip connections every 2 convolutional layers. In 2017, Ioannou et al. [9] adapted the NiN architecture to use 2 to 16 convolutional groups per layer for classifying the CIFAR-10 dataset.

A grouped convolution separates input channels and filters into groups. Each filter processes only input channels entering its group. Each group of filters can be understood as an independent (parallel) path for information flow. This aspect drastically reduces the number of weights in each filter and, therefore, reduces the number of floating-point computations. Grouping 3 × 3 and 5 × 5 spatial convolutions, Ioannou et al. were able to decrease the number of parameters by more than 50% while keeping the NiN classification accuracy. Ioannou et al. also adapted the Resnet-50, Resnet-200, and GoogleLeNet architectures applying 2 to 64 groups per layer when classifying the ImageNet dataset, obtaining parameter reduction while maintaining or improving the classification accuracy. Also in 2017, an improvement for the ResNet architecture called ResNeXt [10] was introduced, replacing the spatial convolutions with parallel paths (groups), reducing the number of parameters.

Several studies have also reported the creation of parameter-efficient architectures with grouped convolutions [11,12,13,14,15]. In 2019, Mingxing Tan et al. [16] developed the EfficientNet architecture. At that time, their EfficientNet-B7 variant was 8.4 times more parameter-efficient and 6.1 times faster than the best existing architecture, achieving 84.3% top-1 accuracy on ImageNet. More than 90% of the parameters of EfficientNets come from standard pointwise convolutions. This aspect opens an opportunity for a huge reduction in several parameters and floating-point operations, as we have exploited in the present paper.

Most parameters in DCNNs are redundant [17,18,19,20,21]. Pruning methods remove connections and neurons found to be irrelevant by different techniques. After training the original network with the full set of connections, the removal is carried out [22,23,24,25,26,27]. Our method differs from pruning as we reduce the number of connections before the training starts, while pruning does after training. Therefore, our method can save computing resources during training time.

In previous works [28,29], we proposed replacing standard pointwise convolutions with a sub-architecture that contains two grouped pointwise convolutional layers (K and L), an interleaving layer that mixes channels from layer K before feeding the layer L, and a summation at the end that sums the results from both convolutional layers. Our original method accepts a hyperparameter Ch, which denotes the number of input channels fed to each group of filters. Then, our method computes the number of groups of filters and filters per group according to the division of original input channels and filters by Ch. Our original method avoided substituting the layers where the divisions were not exact.

In this paper, we propose an enhanced scheme to allow computing the number of groups in a flexible manner, in the sense that the divisibility constraints do not have to be considered anymore. By applying our method to all pointwise convolutional layers of an EfficientNet-B0 architecture, we are able to reduce a huge amount of resources (trainable parameters, floating-point computations) while maintaining the learning capacity.

This paper is structured as follows: Section 2 details our improved solution for grouping pointwise convolutions while skipping the constraints of divisibility found in our previous method. Section 3 details the experiments carried out for testing our solution. Section 4 summarizes the conclusions and limitations of our proposal.

## 2. Methodology

### 2.1. Mathematical Ground for Regular Pointwise Convolutions

Let $X^{i}=\left\{x_{1}^{i},x_{2}^{i},\ldots ,x_{Ic_{i}}^{i}\right\}$ be a set of input feature maps (2D lattices) for a convolutional layer *i* in a DCNN, where $Ic_{i}$ denotes the number of input channels for this layer. Let $W^{i}=\left\{w_{1}^{i},w_{2}^{i},\ldots ,w_{F_{i}}^{i}\;\right\}$ be a set of filters containing the weights for convolutions, where $F_{i}$ denotes the number of filters at layer *i*, which is also the number of output channels of this layer. Following the notation proposed in [30], a regular DCNN convolution can be mathematically expressed as in Equation (Equation 1):(1)$$\begin{array}{cc} X^{i+1} & =\;W^{i}\;⨂\;X^{i}\\ \; & =\;\{w_{1}^{i}*X^{i},\;w_{2}^{i}*X^{i},\ldots ,w_{F_{i}}^{i}*X^{i}\} \end{array}$$
where the ⨂ operator indicates that filters in $W^{i}$ are convolved with feature maps in $X^{i}$, using the ∗ operator to indicate a 3D tensor multiplication and shifting of a filter $w_{j}^{i}$ across all patches of the size of the filter in all feature maps. For simplicity, we are ignoring the bias terms. Consequently, $X^{i+1}$ will contain $F_{i}$ feature maps that will feed the next layer i+1. The tensor shapes of involved elements are the following:(2)$$\begin{array}{cc} \; & X^{i}\in R^{H\times W\times {Ic}_{i}}\\ \; & W^{i}\in R^{F_{i}\times S\times S\times {Ic}_{i}}\;\to \;\;w_{j}^{i}\in R^{S\times S\times {Ic}_{i}}\;\\ \; & X^{i+1}\in R^{H\times W\times F_{i}} \end{array}$$
where H×W is the size (height, width) of feature maps, and S×S is the size of a filter (usually square). In this paper we work with S=1 because we are focused on pointwise convolutions. In this case, each filter $w_{j}^{i}$ carries $Ic_{i}$ weights. The total number of weights $P_{i}$ in layer *i* is obtained with a simple multiplication:(3)$$P_{i}=Ic_{i}\cdot F_{i}$$

### 2.2. Definition of Grouped Pointwise Convolutions

For expressing a grouped pointwise convolution, let us split the input feature maps and the set of filters in $G_{i}$ groups, as $X^{i}=\left\{X_{1}^{i},X_{2}^{i},\ldots ,X_{G_{i}}^{i}\right\}$ and $W^{i}=\left\{W_{1}^{i},W_{2}^{i},\ldots ,W_{G_{i}}^{i}\right\}$. Assuming that both $Ic_{i}$ and $F_{i}$ are divisible by $G_{i}$, the elements in $X^{i}$ and $W^{i}$ can be evenly distributed through all their subset $X_{j}^{i}$ and $W_{j}^{i}$. Then, Equation (Equation 1) can be reformulated as Equation (Equation 4):(4)$$X^{i+1}\;=\;\left\{W_{1}^{i}⊗X_{1}^{i},W_{2}^{i}⊗X_{2}^{i},\ldots ,W_{G_{i}}^{i}⊗X_{G_{i}}^{i}\right\}$$

The shapes of the subsets are the following:(5)$$\begin{array}{cc} \; & X_{m}^{i}\in R^{H\times W\times \frac{{Ic}_{i}}{G_{i}}}\;\\ \; & W_{m}^{i}\in R^{{Fg}_{i}\times 1\times 1\times \frac{{Ic}_{i}}{G_{i}}}\;\to \;w_{j}^{i,m}\in R^{1\times 1\times \frac{{Ic}_{i}}{G_{i}}} \end{array}$$
where ${Fg}_{i}$ is the number of filters per group, namely, Fgi=FiGi. Since each filter $w_{j}^{i,m}$ only convolves on a fraction of input channels (IciGi), the total number of weights per subset $W_{m}^{i}$ is FiGi·IciGi. Multiplying the last expression by the number of groups provides the total number of weights $\bar{P_{i}}$ in a grouped pointwise convolutional layer *i*:(6)$$\bar{P_{i}}\;=\;\left(Ic_{i}\cdot F_{i}\right)/G_{i}$$
Equation (Equation 6) shows that the number of trainable parameters is inversely proportional to the number of groups. However, grouping has the evident drawback that it prevents the filters to be connected with all input channel, which reduces the possible connections of input channels for learning new patterns. As it may lead to a lower learning capacity of the DCNN, one must be cautious with using such grouping technique.

### 2.3. Improved Scheme for Reducing the Complexity of Pointwise Convolutions

Two major limitations of our previous method were inherited from constraints found in most deep learning APIs:The number of input channels $Ic_{i}$ must be multiple of the number of groups $G_{i}$.The number of filters $F_{i}$ must be multiple of the number of groups $G_{i}$.

The present work circumvents the first limitation by replicating channels from the input. The second limitation is circumvented by adding a second parallel path with another pointwise grouped convolution when required. Figure 1 shows an example of our updated architecture.

Details of this process are described below, which is applied to substitute each pointwise convolutional layer *i* found in the original architecture. To explain the method, we start detailing the construction of the layer K shown in Figure 1. For simplicity, we drop the index *i* and use the index *K* to refer to the original hyperparameters, i.e., we use $Ic_{K}$ instead of $Ic_{i}$, $F_{K}$ instead of $F_{i}$. Also, we will use the indexes K1 and K2 to refer the parameters of the two parallel paths that may exist in layer K.

First of all, we must manually specify the value of the hyperparameter Ch. In the graphical example shown in Figure 1, we set Ch=4. The rest of hyperparameters such as number of groups in layers K and L are determined automatically by the rules of our algorithm, according to the chosen value of Ch, the number of input channels $Ic_{K}$ and the number of filters $F_{K}$. We do not have a procedure to find the optimal value of Ch, hence we must apply ablation studies on a range of Ch values as shown in the results section. For the example in Figure 1, we have chosen the value of Ch to obtain a full variety of situations that must be tackled by our algorithm, i.e., non-divisibility conditions.

### 2.4. Definition of Layer K

The first step of the algorithm is to compute the number of groups in branch K1, as in Equation (Equation 7):(7)$$G_{K1}\;=\;\left\lceil \frac{{Ic}_{K}}{Ch}\right\rceil$$

Since the number of input channels $Ic_{K}$ may not be divisible by Ch, we use the ceiling operator on the division to obtain an integer number of groups. In the example, $G_{K1}=\left\lceil 14/4\right\rceil =4$. Thus, the output of filters in branch K1 can be defined as in (Equation 8):(8)$$K1=\left\{W_{1}^{K1}⊗X_{1}^{K},W_{2}^{K1}⊗X_{2}^{K},\ldots ,W_{G_{K1}}^{K1}⊗X_{G_{K1}}^{K}\right\}$$

The subsets $X_{m}^{K}$ are composed of input feature maps $x_{j}$, collected in a sorted manner, i.e., $X_{1}^{K}=\left\{x_{1},\;x_{2},\ldots ,x_{Ch}\right\}$, $X_{2}^{K}=\left\{x_{Ch+1},\;x_{Ch+2},\ldots ,x_{2Ch}\right\}$, etc. Equation (Equation 9) provides a general definition of which feature maps $x_{j}$ are included in any feature subset $X_{m}^{K}$:(9)$$X_{m}^{K}=\left\{x_{a+1},\;x_{a+2},\ldots ,x_{a+Ch}\right\},\;a=\left(m-1\right)\cdot \;Ch$$

However, if $Ic_{K}$ is not divisible by Ch, the last group $m=G_{K1}$ would not have Ch channels. In this case, the method will complete this last group replicating Ch−b initial input channels, where *b* is computed as stated in Equation (Equation 10):(10)$$\begin{array}{cc} X_{G_{K1}}^{K} & =\left\{x_{a+1},\;x_{a+2},\ldots ,x_{a+b},x_{1},x_{2},\ldots ,\;x_{Ch-b}\right\},\\ a & =\left(G_{K1}-1\right)\cdot \;Ch,\\ b & =G_{K1}\cdot \;Ch-{Ic}_{K} \end{array}$$

It can be proved that *b* will always be less or equal than Ch, since *b* is the excess of the integer division $Ic_{K}/Ch$, i.e., $G_{K1}\cdot \;Ch$ will always be above or equal to $Ic_{K}$, but less than $Ic_{K}+Ch$, because otherwise $G_{K1}$ would increase its value (as a quotient of $Ic_{K}/Ch$). In the example, b=2, hence $X_{4}^{K1}=\left\{x_{13},x_{14},x_{1},x_{2}\right\}$.

Then, the method calculates the number of filters per group ${Fg}_{K1}$ as in (Equation 11):(11)$${Fg}_{K1}\;=\;\left\lfloor \frac{F_{K}}{G_{K1}}\right\rfloor$$

To avoid divisibility conflicts, this time we have chosen the floor integer division. For the first path K1, each of the filter subsets shown in (Equation 8) will contain the following filters:(12)$$\begin{array}{cc} W_{m}^{K1}= & \left\{w_{1}^{K1,m},\;w_{2}^{K1,m},\ldots ,w_{{Fg}_{K1}}^{K1,m}\right\}\\ \; & w_{j}^{K1,m}\in R^{1\times 1\times Ch} \end{array}$$

For the first path of the example, the number of filters per group is ${Fg}_{K1}=10/4=2$. So, the first path has 4 groups ($G_{K1}$) of 2 filters (${Fg}_{K1}$), each filter being connected to 4 input channels (Ch).

If $F_{K}$ is not divisible by Ch, a second path K2 will provide as many groups as filters not provided in K1, with one filter per group, to complete the total number of filters $F_{K}$:(13)$$\begin{array}{cc} G_{K2}\; & =\;F_{K}-{Fg}_{K1}\cdot \;G_{K1}\\ {Fg}_{K2}\; & =\;1 \end{array}$$

In the example, $G_{K2}=2$. The required input channels for the second path is $Ch\cdot G_{K2}$. The method obtains those channels reusing the same subsets of input feature maps $X_{m}^{K}$ shown in (Equation 9). Hence, the output of filters in path K2 can be defined as in (Equation 14):(14)$$K2=\left\{w_{1}^{K2}*X_{1}^{K},w_{2}^{K2}*X_{2}^{K},\ldots ,w_{G_{K2}}^{K2}*X_{G_{K2}}^{K}\right\}$$
where ${w_{j}^{K2}\in R}^{1\times 1\times Ch}$. Therefore, each filter in K2 operates on exactly the same subset of input channels than the corresponding subset of filters in K1. Hence, each filter in the second path can be considered as belonging to one of the groups of the first path.

It must be noticed that $G_{K2}$ will always be less than $G_{K1}$. This is true because $G_{K2}$ is the reminder of the integer division $F_{K}/G_{K1}$, as can be deduced from (Equation 11) and (Equation 13). This property warranties that there will be enough subsets $X_{m}^{K}$ for this second path.

After defining paths K1 and K2 in layer K, the output of this layer is the concatenation of both paths:(15)$$K=\left\{K1,\;K2\right\}$$

The total number of channels after the concatenation is equal to $F_{K}=G_{K1}\cdot {Fg}_{K1}+G_{K2}$.

### 2.5. Interleaving Stage

As mentioned above, grouped convolutions inherently face a limitation: each parallel group of filters computes its output from their own subset of input channels, preventing combinations of channels connected to different groups. To alleviate this limitation, we propose to interleave the output channels from the convolutional layer K.

The interleaving process simply consists in arranging the odd channels first and the even channels last, as noted in Equation (Equation 16):(16)$$\begin{array}{cc} I^{K}= & \{k_{1},k_{3},k_{5},\ldots ,k_{2c-1},\\ & \;k_{2},k_{4},k_{6},\ldots ,k_{2c}\}\\ c= & F_{K}/2 \end{array}$$

Here we are assuming that $F_{K}$ is even. Otherwise, the list of odd channels will include an extra channel $k_{2c+1}$.

### 2.6. Definition of Layer L

The interleaved output feeds the grouped convolutions in layer L to process data coming from more than one group from the preceding layer K.

To create layer L, we apply the same algorithm as for layer K, but now the number of input channels is equal to $F_{K}$ instead of $Ic_{K}$.

The number of groups in path L1 is computed as:(17)$$G_{L1}\;=\;\left\lceil \frac{F_{K}}{Ch}\right\rceil$$

Note that $G_{L1}$ may not be equal to $G_{K1}$. In the example, $G_{L1}=10/4=3$.

Then, the output of L1 is computed as in (Equation 18), where the input channel groups $I_{m}^{K}$ come from the interleaving stage. Each group is composed of Ch channels, whose indexes are generically defined in (Equation 19):(18)$$L1=\;\left\{W_{1}^{L1}⊗I_{1}^{K},W_{2}^{L1}⊗I_{2}^{K},\ldots ,W_{G_{L1}}^{K1}⊗I_{G_{L1}}^{K}\right\}$$
(19)$$\begin{array}{cc} I_{m}^{K} & =\left\{i_{a+1}^{K},\;i_{a+2}^{K},\ldots ,i_{a+Ch}^{K}\right\},\\ a & =(m-1)\cdot \;Ch \end{array}$$

Again, the last group of indexes may not contain Ch channels due to a non-exact division condition in (Equation 17). Similar to path K1, for path L1 the missing channels in the last group will be supplied by replicating Ch−b initial interleaved channels, where *b* is computed as stated in Equation (Equation 20):(20)$$\begin{array}{cc} I_{G_{L1}}^{K} & =\left\{i_{a+1}^{K},\;i_{a+2}^{K},\ldots ,i_{a+b}^{K},i_{1}^{K},i_{2}^{K},\ldots ,\;i_{Ch-b}^{K}\right\},\\ a & =\left(G_{L1}-1\right)\cdot \;Ch,\\ b & =G_{L1}\cdot \;Ch\;-F_{K} \end{array}$$

The number of filters per group ${Fg}_{L1}$ is computed as in (Equation 21):(21)$${Fg}_{L1}\;=\;\left\lfloor \frac{F_{K}}{G_{L1}}\right\rfloor$$

In the example, ${Fg}_{L1}=10/3=3$. Each group of filters $W_{m}^{L1}$ shown in (Equation 18) can be defined as in (Equation 22), each one containing ${Fg}_{L1}$ convolutional filters of Ch inputs:(22)$$\begin{array}{cc} W_{m}^{L1}= & \left\{w_{1}^{L1,m},\;w_{2}^{L1,m},\ldots ,w_{{Fg}_{L1}}^{L1,m}\right\}\\ \; & w_{j}^{L1,m}\in R^{1\times 1\times Ch} \end{array}$$

It should be noted that if the division in (Equation 21) is not exact, the number of output channels from layer L may not reach the required $F_{K}$ outputs. In this case, a second path L2 will be added, with the following parameters:(23)$$\begin{array}{cc} G_{L2}\; & =\;F_{K}-{Fg}_{L1}\cdot \;G_{L1}\\ {Fg}_{L2}\; & =\;1 \end{array}$$

In the example, $G_{L2}=1$. The output of path L2 is computed as in (Equation 24), defining one extra convolutional filter for some initial groups of interleaved channels declared in (Equation 18) and (Equation 19), taking into account that $G_{L2}$ will always be less than $G_{L1}$ according to the same reasoning done for $G_{K2}$ and $G_{K1}$:(24)$$L2=\;\left\{w_{1}^{L2}*I_{1}^{K},w_{2}^{L2}*I_{2}^{K},\ldots ,w_{G_{L2}}^{L2}*I_{G_{L2}}^{K}\right\}$$

The last step in defining the output of layer L is to join the outputs of paths L1 and L2:(25)$$L=\;\left\{L1,\;L2\right\}$$

### 2.7. Joining of Layers

Finally, the output of both convolutional layers K and L are summed to create the output of the original layer:(26)$$X^{i+1}=\;K+L$$

Compared to concatenation, summation has the advantage of allowing a residual learning in the filters of layer L, because gradient can be backpropagated through L filters or directly to K filters. In other words, residual layers provide more learning capacity with low degree of downsides due to increasing the number of layers (i.e., overfitting, longer training time, etc.) In the results section, we present an ablation study that contains experiments done without the interleaving and the L layers (rows labeled with “no L”). These experiments empirically prove that the interleaving mechanism and the secondary L layer help in improving the sub-architecture accuracy, with low impact.

It is worth mentioning that we only add the layer L an the interleaving when the number of input channels is bigger or equal to the number of filters in layer K.

### 2.8. Computing the Number of Parameters

We can compute the total number of parameters of our sub-architecture. First, Equation (Equation 27) shows that the number of filters in layer K is equal to the number of filters in layer L, which in turn is equal to the total number of filters in the original convolutional layer $F_{i}$:(27)$${Fg}_{K1}\cdot \;G_{K1}+G_{K2}=\;{Fg}_{L1}\cdot \;G_{L1}+G_{L2}=F_{i}$$

Then, the total number of parameters $\bar{\bar{P_{i}}}$ is twice the number of original filters multiplied by the number of input channels per filter:(28)$$\bar{\bar{P_{i}}}\;=\;2\left(F_{i}\cdot Ch\right)$$

Therefore, comparing Equation (Equation 28) with (Equation 3), it is clear that Ch must be significantly less than $Ic_{i}/2$ to reduce the number of parameters of a regular pointwise convolutional layer. Also, comparing Equation (Equation 28) with (Equation 6), our sub-architecture provides a parameter reduction similar to a plain grouped convolutional layer when Ch is around $Ic_{i}/2\cdot G_{i}$, although we cannot specify a general $G_{i}$ term because of the complexity of our pair of layers with possibly two paths per layer.

The requirement for a low value of Ch is also necessary to ensure that divisions in Equations (Equation 7) and (Equation 17) provide quotients above one, otherwise our method will not create grouping. Hence, Ch must be less or equal to $Ic_{i}/2$ and $F_{i}/2$. These are the only two constraints that our method is restricted by.

As shown in Table 1, pointwise convolutional layers found in real networks such as EfficientNet-B0 have significant Figures for $Ic_{i}$ and $F_{i}$, either hundreds or thousands. Therefore, values of Ch less or equal than 32 will ensure a good ratio of parameter reduction for most of these pointwise convolutional layers.

EfficientNet is one of the most complex (but efficient) architectures that can be found in the literature. To our method, the degree of complexity of a DCNN is mainly related to the maximum number of input channels and output features in any pointwise convolutional layer. Our method does not care about the number of layers, neither in depth nor in parallel, because it works on each layer independently. Therefore, the degree of complexity of EfficientNet-B0 can be considered significantly high, taking into account the values shown in the last row of Table 1. Arguably, other versions of EfficientNet (B1, B2, etc.) and other types of DCNN can exceed those values. In such cases, higher values of Ch may be necessary, but we cannot provide any rule to forecast its optimum value for the configuration of any pointwise convolutional layer.

### 2.9. Activation Function

In 2018, Prajit et al. [31] tested a number of activation functions. In their experimentation, they found that the best performing one was the so-called “swish”, shown in Equation (Equation 29).
(29)f(x)=x·sigmoid(βx)

In previous works [28,29], we used the ReLU activation function. In this work, we use the swish activation function. This change gives us better results in our ablation experiments shown on Table 5.

### 2.10. Implementation Details

We tested our optimization by replacing original pointwise convolutions in the EfficientNet-B0 and named it as “kEffNet-B0 V2”. With CIFAR-10, we tested an additional modification that skips the first 4 convolutional strides, allowing input images with 32 × 32 pixels instead of the original resolution of 224 × 224 pixels.

In all our experiments, we saved the trained network from the epoch that achieved the lowest validation loss for testing with the test dataset. Convolutional layers are initialized with Glorot’s method [32]. All experiments were trained with RMSProp optimizer, data augmentation [33] and cyclical learning rate schedule [34]. We worked with various configurations of hardware with NVIDIA video cards. Regarding software, we did our experiments with K-CAI [35] and Keras [36] on the top of Tensorflow [37].

Our source code is publicly available at: https://github.com/joaopauloschuler/kEffNetV2/, accessed on 1 September 2022.

### 2.11. Horizontal Flip

In some experiments, we run the model twice with the input image and its horizontally flipped version. The output from the softmax from both runs is summed before class prediction. In these experiments, the number of floating-point computations doubles, although the number of trainable parameters remains the same.

## 3. Results and Discussion

In this section, we present and discuss the results of the proposed scheme with three image classification datasets: CIFAR-10 dataset [38], Malaria dataset, and colorectal cancer histology dataset [39,40].

### 3.1. Results on the CIFAR-10 Dataset

The CIFAR-10 dataset [38] is a subset of [41] and consists of 60k 32 × 32 images belonging to 10 different classes: airplane, automobile, bird, cat, deer, dog, frog, horse ship and truck. These images are taken from natural and uncontrolled lightning environment. They contain only one prominent instance of the object to which the class refers to. The object may be partially occluded or seen from an unusual viewpoint. This dataset has 50k images for training and 10k images for test. We picked 5k images for validation and left the training set with 45k images. We run experiments with 50 and 180 epochs.

On Table 2 we compare kEffNet-B0 V1 (our previous method) and V2 (our current method), for two values of Ch. We can see that our V2 models has slightly more reduction in both number of parameters and floating-point computations than the V1 counterpart models, while achieving slightly higher accuracy. Specifically, V2 models save 10% of the parameters (from 1,059,202 to 950,650) and 11% of the floating-point computations (from 138,410,206 to 123,209,110) of V1 models. All of our variants obtain similar accuracy to the baseline with a remarkable reduction of resources (at least 26.3% of trainable parameters and 35.5% of computations).

As the scope of this work is limited to small datasets and small architectures, we only experimented with the smallest EfficientNet variant (EfficientNet-B0) and our modified variant (kEffNet-B0). Nevertheless, Table 3 provides the number of trainable parameters of the other EfficientNet variants (original and parameter-reduced). Equation (Equation 3) indicates that the number of parameters grows with the number of filters and the number of input channels. Equation (Equation 6) indicates that the number of parameters decreases with the number of groups. As we create more groups when the number of input channels grows, we expect to find bigger parameter savings on larger models. This saving can be seen on Table 3.

We also tested our kEffNet-B0 with 2, 4, 8, 16 and 32 channels per group for 50 epochs as shown in Table 4. As expected, the test classification accuracy increases when allocating more channels per group: from 84.26% for Ch = 2 to 93.67% for Ch = 32. Also, the resource saving decreases as the number of channels per group increase: from 7.8% of parameters and 11.4% of computations for Ch = 2 to 23.6% of parameters and 31.6% of computations for Ch = 32 (compared to the baseline). For CIFAR-10, if we aim to achieve an accuracy comparable to the baseline, we must choose at least 16 channels per group. If we add an extra run per image sample with horizontal flipping when training kEffNet-B0 V2 32ch, the classification accuracy increases from 93.67% to 94.01%.

Table 5 replicates most of the results shown in Table 4, but comparing the effect of not including layer L and interleaving, and also substituting the swish activation function with the typical ReLU. As can be observed, disabling layer L has a noticeable degradation on test accuracy when the values of Ch are smaller. For example, when Ch = 4, the performance drops more than 5%. On the other hand, when Ch = 32 the drop is less than 0.5%. This is logical taking into account that, the more channels are included per group, the more chances are to combine input features in the filters. Therefore, a second layer and the corresponding interleaving is not as crucial as when the filters of layer K are fed with fewer channels.

In the comparison of activation functions, the same effect can be appreciated: the swish function works better than the ReLU function, but provides less improvement for larger number of channels per group. Nevertheless, the gain in the least difference case (32 ch) is still profitable, with more than 1.5% of extra test accuracy when using the swish activation function.

Table 6 shows the effect in accuracy when classifying the CIFAR-10 dataset with EfficientNet-B0 and our kEffNet-B0 V2 32ch variant for 180 epochs instead of 50 epochs. The additional training epochs assign slightly higher test accuracy to the baseline than to our core variant. When adding horizontal flipping, our variant has slightly surpassed the baseline results. Nevertheless, all three results can be considered similar to each other, but our variant offers a significant saving in parameters and computations. Although the H flipping doubles the computational cost of our core variant, it still remains only a fraction (63.3%) of the baseline computational cost.

### 3.2. Results on the Malaria Dataset

The Malaria dataset [40] has 27,558 cell images from infected and healthy cells separated into 2 classes. There is the same number of images for healthy and infected cells. From the original 27,558 images set, we separated 10% of the images (2756 images) for validation and another 10% for testing. On all training, validation, and test subsets, there are 50% of healthy cell images. We quadruplicated the number of validation images by flipping these images horizontally and vertically, resulting in 11,024 images for validation.

On this dataset, we tested our kEffNet-B0 with 2, 4, 8, 12, 16, and 32 channels per group, as well as the baseline architecture, as shown in Table 7. Our variants have from 7.5% to 23.5% of the trainable parameters and from 15.7% to 42.2% of the computations allocated by the baseline architecture. Although the worst classification accuracy was found with the smallest variant (2ch), its classification accuracy is less than 1% inferior to the best performing variant (16ch) and only 0.69% below the baseline performance. With only 8 channels per group, our method equals the baseline accuracy with a small portion of the parameters (10.8%) and computations (22.5%) required by the baseline architecture. Curiously, our 32ch variant is slightly worse than the 16ch variant, but still better than the baseline. It is an example that a rather low complexity of the input images may require less channels per filter (and more parallel groups of filters), to optimally capture the relevant features of images.

### 3.3. Results on the Colorectal Cancer Histology Dataset

The collection of samples in colorectal cancer histology dataset [39] contains 5000 150 × 150 images separated into 8 classes: adipose, complex, debris, empty, lympho, mucosa, stroma, and tumor. Similar to what we did with the Malaria dataset, we separated 10% of the images for validation and another 10% for testing. We also quadruplicated the number of validation images by flipping these images horizontally and vertically.

On this dataset, we tested our kEffNet-B0 with 2, 4, 8, 12, and 16 channels per group, as well as the baseline architecture, as shown in Table 8. Similar to the Malaria dataset, higher values of channels per group do not lead to better performance. In this case, the variants with the highest classification accuracy are 4ch and 8ch, achieving 98.02% of classification accuracy, outperforming the baseline accuracy in 0.41%. The 16ch variant has obtained the same accuracy than the 2ch variant, but doubling the required resources. Again, it indicates that the complexity of the images plays a role in the selection of the optimal number of channels per group. In other words, simpler images may require less channels per group. Unfortunately, the only method we know to find out this optimal value is performing theses scanning experiments.

## 4. Conclusions and Future Work

This paper presented an efficient scheme for decreasing the complexity of pointwise convolutions in DCNNs for image classification based on interleaved grouped filters with no divisibility constraints. From our experiments, we can conclude that connecting all input channels from the previous layer to all filters is unnecessary: grouped convolutional filters can achieve the same learning power with a small fraction of resources (1/3 of floating-point computations, 1/4 of parameters). Our enhanced scheme avoids the divisibility contraints, furter reducing the required resources (up to 10% less) while maintaining or slightly surpassing the accuracy of our previous method.

We have made ablation studies to obtain the optimal number of channels per group for each dataset. For colorectal cancer dataset, this number is surprisingly low (4 channels per group). On the other side, for CIFAR-10 the best results require at least 16 channels per group. This fact indicates that the complexity of the input images affects the optimal configuration of our sub-architecture.

As the main limitation of our method, it cannot determine the optimal number of channels per group automatically, according to the complexity of each pointwise convolutional layer to be substituted and the complexity of input images. A second limitation is that the same number of channels per group is applied to all pointwise convolutional layers of the target architecture, regardless of the specific complexity of each layer. This limitation could be easily tackled by setting Ch as a fraction of the total number of parameters of each layer. This is a straightforward task for future research. Besides, we will apply our method to different problems, such as instance and semantic image segmentation, developing an efficient deep learning-based seismic acoustic impedance inversion method [42], object detection, and forecasting.

## Figures and Tables

**Figure 1 entropy-24-01264-f001:**
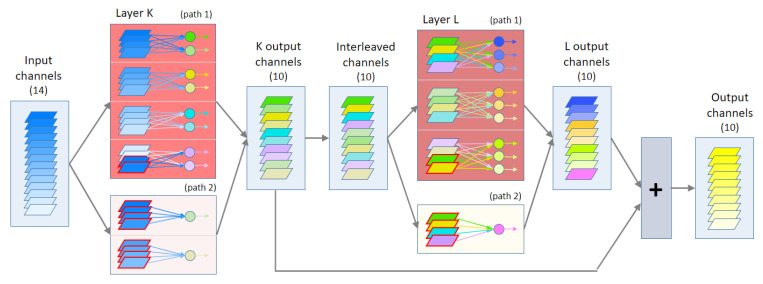
A schematic diagram of our pointwise convolution replacement. This example replaces a pointwise convolution with 14 input channels and 10 filters. It contains two convolutional layers, K and L, one interleaving, and one summation layer. Channels surrounded by a red border represent replicated channels.

**Table 1 entropy-24-01264-t001:** For a standard pointwise convolution with Ic input channels, *F* filters, *P* parameters and a given number of channels per group Ch, this Table shows the calculated parameters for layers K and L: the number of groups $G_{<layer><path>}$ and the number of filters per group $Fg_{<layer><path>}$. The last 2 columns show the total number of parameters and its percentage with respect to the original layer.

Original Settings				Layer K			Layer L			K+L Params	
Ic	F	P	Ch	$G_{K1}$	${\mathrm{Fg}}_{K1}$	$G_{K2}$	$G_{L1}$	${\mathrm{Fg}}_{L1}$	$G_{L2}$	**Total**	**%**
14	10	140	4	4	2	2	3	3	1	80	57.14%
160	3840	614,400	16	10	384	0	0	0	0	61,440	10.00%
			32	5	768	0	0	0	0	122,880	20.00%
192	1152	221,184	16	12	96	0	0	0	0	18,432	8.33%
			32	6	192	0	0	0	0	36,864	16.67%
1152	320	368,640	16	72	4	32	20	16	0	10,240	2.78%
			32	36	8	32	10	32	0	20,480	5.56%
3840	640	2,457,600	16	240	2	160	40	16	0	20,480	0.83%
			32	120	5	40	20	32	0	40,960	1.67%

**Table 2 entropy-24-01264-t002:** Comparing EfficientNet-B0, kEffNet-B0 V1 and kEffNet-B0 V2 with CIFAR-10 dataset after 50 epochs.

Model	Parameters	%	Computations	%	Test acc.
EfficientNet-B0 baseline	4,020,358	100.0%	389,969,098	100.0%	93.33%
kEffNet-B0 V1 16ch	639,702	15.9%	84,833,890	21.8%	92.46%
kEffNet-B0 V2 16ch	623,226	15.5%	82,804,374	21.2%	92.61%
kEffNet-B0 V1 32ch	1,059,202	26.3%	138,410,206	35.5%	93.61%
kEffNet-B0 V2 32ch	950,650	23.6%	123,209,110	31.6%	93.67%

**Table 3 entropy-24-01264-t003:** Number of trainable parameters for EfficientNet, kEffNet V2 16ch and kEffNet V2 32ch with a 10 classes dataset.

Variant	EfficientNet	kEffNet V2 16ch	%	kEffNet V2 32ch	%
B0	4,020,358	623,226	15.50%	950,650	23.65%
B1	6,525,994	968,710	14.84%	1,389,062	21.29%
B2	7,715,084	983,198	12.74%	1,524,590	19.76%
B3	10,711,602	1,280,612	11.96%	2,001,430	18.68%
B4	17,566,546	1,858,440	10.58%	2,911,052	16.57%
B5	28,361,274	2,538,870	8.95%	4,011,626	14.14%
B6	40,758,754	3,324,654	8.16%	5,245,140	12.87%
B7	63,812,570	4,585,154	7.19%	7,254,626	11.37%

**Table 4 entropy-24-01264-t004:** Ablation study done with the CIFAR-10 dataset for 50 epochs, comparing the effect of varying the number of channels per group. It also includes the improvement achieved by double training kEffNet-B0 V2 32ch with original images and horizontally flipped images.

Model	Parameters	%	Computations	%	Test acc.
EfficientNet-B0 baseline	4,020,358	100.0%	389,969,098	100.0%	93.33%
kEffNet-B0 V2 2ch	311,994	7.8%	44,523,286	11.4%	84.36%
kEffNet-B0 V2 4ch	354,818	8.8%	49,487,886	12.7%	87.66%
kEffNet-B0 V2 8ch	444,346	11.1%	60,313,526	15.5%	90.53%
kEffNet-B0 V2 16ch	623,226	15.5%	82,804,374	21.2%	92.61%
kEffNet-B0 V2 32ch	950,650	23.6%	123,209,110	31.6%	93.67%
kEffNet-B0 V2 32ch + H Flip	950,650	23.6%	246,418,220	63.3%	94.01%

**Table 5 entropy-24-01264-t005:** Extra experiments made for kEffNet-B0 V2 4ch, 8ch, 16ch and 32ch variants. Rows labeled with “no L” indicate experiments done using only layer K, i.e., disabling layer L and the interleaving. Rows labeled with “ReLU” replace the swish activation function by ReLU.

Model	Parameters	%	Computations	%	Test acc.
EfficientNet-B0 baseline	4,020,358	100.0%	389,969,098	100.0%	93.33%
kEffNet-B0 V2 4ch	354,818	8.8%	49,487,886	12.7%	87.66%
kEffNet-B0 V2 4ch no L	342,070	8.5%	48,064,098	12.3%	82.44%
kEffNet-B0 V2 4ch ReLU	354,818	8.8%	47,595,914	12.2%	85.34%
kEffNet-B0 V2 8ch	444,346	11.1%	60,313,526	15.5%	90.53%
kEffNet-B0 V2 8ch no L	422,886	10.5%	57,466,370	14.7%	89.27%
kEffNet-B0 V2 8ch ReLU	444,346	11.1%	58,421,554	15.0%	88.82%
kEffNet-B0 V2 16ch	623,226	15.5%	82,804,374	21.2%	92.61%
kEffNet-B0 V2 16ch no L	584,934	14.6%	77,356,802	19.8%	91.52%
kEffNet-B0 V2 16ch ReLU	623,226	15.5%	80,912,406	20.8%	91.16%
kEffNet-B0 V2 32ch	950,650	23.6%	123,209,110	31.6%	93.67%
kEffNet-B0 V2 32ch no L	879,750	21.9%	112,684,706	28.9%	93.21%
kEffNet-B0 V2 32ch ReLU	950,650	23.7%	121,317,142	31.1%	92.00%

**Table 6 entropy-24-01264-t006:** Results obtained with the CIFAR-10 dataset after 180 epochs.

Model	Parameters	%	Computations	%	Test acc.
EfficientNet-B0 baseline	4,020,358	100.0%	389,969,098	100.0%	94.86%
kEffNet-B0 V2 32ch	950,650	23.6%	123,209,110	31.6%	94.45%
kEffNet-B0 V2 32ch + H Flip	950,650	23.6%	246,418,220	63.3%	94.95%

**Table 7 entropy-24-01264-t007:** Results obtained with the Malaria dataset after 75 epochs.

Model	Parameters	%	Computations	%	Test acc.
EfficientNet-B0 baseline	4,010,110	100.0%	389,958,834	100.0%	97.39%
kEffNet-B0 V2 2ch	301,746	7.5%	61,196,070	15.7%	96.70%
kEffNet-B0 V2 4ch	344,570	8.6%	69,691,358	17.9%	96.95%
kEffNet-B0 V2 8ch	434,098	10.8%	87,725,254	22.5%	97.39%
kEffNet-B0 V2 12ch	524,026	13.1%	106,199,566	27.2%	97.31%
kEffNet-B0 V2 16ch	612,978	15.3%	124,672,934	32.0%	97.61%
kEffNet-B0 V2 32ch	940,402	23.5%	164,422,950	42.2%	97.57%

**Table 8 entropy-24-01264-t008:** Results obtained with the colorectal cancer dataset after 1000 epochs.

Model	Parameters	%	Computations	%	Test acc.
EfficientNet-B0 baseline	4,017,796	100.0%	389,966,532	100.0%	97.61%
kEffNet-B0 V2 2ch	355,064	8.8%	61,203,768	15.7%	97.62%
kEffNet-B0 V2 4ch	397,888	9.9%	69,699,056	17.9%	98.02%
kEffNet-B0 V2 8ch	487,416	12.1%	87,732,952	22.5%	98.02%
kEffNet-B0 V2 12ch	531,712	13.2%	106,207,264	27.2%	97.22%
kEffNet-B0 V2 16ch	620,664	15.4%	124,680,632	32.0%	97.62%

## Data Availability

Datasets used in this study are publicly available: CIFAR-10 [38], Colorectal cancer histology [39] and Malaria [40]. Software APIs are also publicly available: K-CAI [35] and Keras [36]. Our source code and raw experiment results are publicly available: https://github.com/joaopauloschuler/kEffNetV2, accessed on 1 September 2022.

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
