# Peer review of "An Enhanced Scheme for Reducing the Complexity of Pointwise Convolutions in CNNs for Image Classification Based on Interleaved Grouped Filters without Divisibility Constraints"

_entropy, 2022, doi:10.3390/e24091264_

Round 1

Reviewer 1 Report

Considering the manuscript with the manuscript: entropy-1855873, entitled " Grouped Pointwise
Convolution With Flexible Number of Channels and Filters per Group", herewith I would like to submit
my comments. This paper is well elucidated and well referenced, nevertheless, some more discussions
need to be added to improve the comprehensiveness of demonstration of the method and some needs
to be removed, revised or edited. Some minor issues need to be addressed or corrected to improve the
general quality and readability of this article. The paper needs to be developed in some parts. So based
on my comments that comes in the following, I proposed the paper to be subjected for moderate
revision.

Best Regards

General Comments:

Is the paper new, technically correct, and relevant?

Yes, the paper is new and technically sounds. Results somehow does support the methodology, but
needed to be more cleared by the author in case of properties of the data.

Is the paper well organized?

The paper is properly organized, good literature review, suitable motivation and clear explanation on
results are positive points to that.

Is the abstract concise?

Yes, but I think it needs to be rephrased after revision to add some comments about any artifacts or
negative points in the method, if exist.

Is the introduction motivating?

Yes, Introduction section is motivating.

Are the methodology, results, and conclusions completely developed?

No, they need to be modified and developed according to the technical comments.

Are there language, mathematics, reference, or style errors?
There is no mathematical, reference or style error

Technical Comments:
Are the codes available for this research? As I found, there is no code available for this study, e. g. in
Github. If the authors could make the codes available, the manuscript could be much better evaluated,
not only for reviewers, but also for possible readers. When it is not possible to upload the code for public
access, such as in Github, could they be provided for reviewer for better assessment of the study?

The study is comprehensive and requires large time to be read carefully and being reviewed. The
theoretical background has been well explained in details, and the experiments and related models are
presented and the algorithm in Fig. 1 is also well presented. I think more explanation about the
optimization of the parameters in Fig. 1 is required.

The result comparison parts are well organized and presented. The display way is good. But
quantitative evaluation is somehow too much that one can get lost in that. I think it would be better that
you add more explanation to that.

The authors should explain what limitations did they find out about the proposed method.

How did you evaluate the final result? How did you consider to finally selection a methodology for
the most complicate dprobelm?

What about when the models are more complex?

The introduction section is a nice one. It is architected very beautifully, while written fully academic
and comprehend. I assume that any change in the introduction section is not necessary, but one of the
important tasks after publishing a study is to increase its chance to be seen by the most possible number
of researchers, so I would like to give two recommendations. First, to get your published study in the list
of searched for papers based on keywords, I propose to increase variety of your keywords. In my
viewpoint, they do not cover the whole topic of the study and are not widely searched words. I propose
to add at least the keyword “data analysis”. Second, one of the methods in the publisher’s website that
brings a publication on to the researchers, is based on the similar publications that they have read before.
So, the more you cite similar publication, the more the chance that the search engine in the publisher
website propose your paper to the researcher. Besides of that, it will also complete your introduction
section. As another advantage, it rises new ideas to the researchers by combining various methods, or
resolving drawback of one seen paper by reading the similar one, or extending the methodology to a fully
automatic one. So, based on these points, I would like to ask to cite to the following similar publication in
the manuscript. The proposed publication is: Shahbazi, A., Soleimani Monfared, M., Thiruchelvam, V., Ka

Fei, T., Babasafari, A.A., (2020). Integration of knowledge-based seismic inversion and sedimentological
investigations for heterogeneous reservoir. Journal of Asian Earth Sciences.

The authors should explain what limitations did they find out about the proposed method.

The abstract focusses mainly on the general problem and ignores the other items of the abstract such
as the methodology, good introduction, results and conclusion.

Best Regard

Reviewer 2 Report

This manuscript is interesting and fits the objectives of the journal. In the end it is a beautiful manuscript; but, before the final decision, it is necessary to review some points:

- Another revision of the English used is necessary, because the sentences are long and confusing in several places.

- The title in its current form is unclear; the referee suggests improving it.

- The introduction section should be revised, removing the title of point 2 and reorganizing this part better by merging it with the introduction paragraph. It seems like a repetition of some equations from the introduction into the materials and methods, so it is best to review it and improve the quality of the introduction.

- In line with these changes, perhaps it is better to improve the other sections as well.

- In addition, the conclusions section needs to be reformulated and shortened by clearly mentioning only the key conclusions of this study.

Reviewer 3 Report

The authors focus on the problem of image/data processing using convolutional euronetworks. In image classification using deep convolutional neural networks (DCNNs), the number of parameters in the point convolutions increases rapidly due to the multiplication of the number of filters with the number of input signals transmitted by the channels that originate from the previous layer. A subnetwork model has been published that replaces the point convolutions with significantly fewer point parameters and fewer floating point calculations while maintaining the learning capacity.
The proposed subnetwork uses grouped point convolutions, where each group processes a portion of the input channels. The proposed solution refines the point algorithm so that the groups can have split filters to handle an indivisible number of input channels, output channels and groups. In this case, the previously published methods overlook this possibility and the proposed method replaces the original point convolution methods.
Thus, the new method reduces the number of floating point operations (by 11%) and trainable parameters ( by 10%) compared to the previous method. The optimization was tested on EfficientNet-B0 as the underlying architecture and a comparative analysis was performed on the CIFAR-10, Colorectal Cancer Histology and Malaria classification datasets.
For the datasets, the proposed method achieved a saving of 76%, 89%, and 91% of the dataset in the number of trainable parameters of EfficientNet-B0 system when optimizing while maintaining its test parameters. classification accuracy.
This paper deals with the theoretical analysis and description of the combinatorial problem. It follows justifiable procedures and established scientific practices. It contains an element of novelty. The text would have benefited from the authors' inclusion of tests not only on test series but on cockney sets of images evaluated manually, by hitherto known methods and by the proposed method. Then they compared these results.
From a formal point of view, I have no major comments.

No plagiarism was found.

Round 2

Reviewer 2 Report

The authors have improved their manuscript respecting the comments made!